# AUGMENTED MEMORY NETWORKS FOR STREAMING-BASED ACTIVE ONE-SHOT LEARNING

## ABSTRACT

One of the major challenges in training deep architectures for predictive tasks is the scarcity and cost of labeled training data. Active Learning (AL) is one way of addressing this challenge. In stream-based AL, observations are continuously made available to the learner that have to decide whether to request a label or to make a prediction. The goal is to reduce the request rate while at the same time maximize prediction performance. In previous research, reinforcement learning has been used for learning the AL request/prediction strategy. In our work, we propose to equip a reinforcement learning process with memory augmented neural networks, to enhance the one-shot capabilities. Moreover, we introduce Class Margin Sampling (CMS) as an extension of the standard margin sampling to the reinforcement learning setting. This strategy aims to reduce training time and improve sample efficiency in the training process. We evaluate the proposed method on a classification task using empirical accuracy of label predictions and percentage of label requests. The results indicates that the proposed method, by making use of the memory augmented networks and CMS in the training process, outperforms existing baselines.

## 1 INTRODUCTION

Traditionally, deep learning architectures have been successfully employed in the data regime where labeled data is abundant – meaning that the cost of obtaining the labels can be ignored in the training process. This includes applications such as large scale object recognition, classification of images, videos or texts, as well as other situations where large labeled data sets are available.

In practice there are many situations where labels are scarce and the cost of obtaining the labels is non-negligible. In AL, this problem has been tackled for settings where, during the training process of the model, minimization of training error and cost of querying for the label a given data point is performed. Typically a set of heuristics are employed, for example by looking at the expected information gain of querying for the label, uncertainty sampling (looking for regions of the data space where there is more uncertainty about the label), exploration-exploitation of the data space, among others. Recent advances in neural architectures for Active Learning have explored the possibilities of learning AL strategy, which means using neural networks for learning *how to* active learn. Models that are capable of learning with only a handful examples is also of great interest. This is generally associated with the one-shot learning problem (Koch et al., 2015).

Leveraging both scarcity of data and labels calls for one-shot active learning. We approach this as a stream-based problem, where a learning agent is confronted with new data in a sequential way, where whenever a data point is received, the model needs to decide on whether to predict the label or to request a label from an oracle - preferably with as few requests as possible. Starting from the work in (Woodward & Finn, 2017) we propose a memory-augmented neural architecture, based on the intuition that gradient based predictive learning should be equipped with memory capabilities (allowing transfer and retention of information) and active learning strategies through reinforcement learning with a reward associated with the actions of requesting labels, or classifying a given point. The main contributions of this work can be summarized as follows: 1) we propose and test an active one-shot learning system equipped with memory augmented architectures, 2) we introduce the novel Class Margin Sampling (CMS), as an extension of standard margin sampling to the reinforcement learning setting, with the goal of improving sampling efficiency for AL.

## 2 RELATED WORK

AL has been extensively studied in the past decade (Settles, 2009) and several heuristics for data selection process have been proposed. Mostly, these selection criteria include uncertainty sampling (Tong & Koller, 2002), query-by-committee (Seung et al., 1992), expected model change (Cai et al., 2017) and expected error reduction (Roy & McCallum, 2001). AL has previously been applied in different domains, such as natural language processing (Zhang et al., 2017; Asghar et al., 2017; Shen et al., 2017; Buck et al., 2017), computer vision (Sener & Savarese, 2018; Wang et al., 2017; Beluch et al., 2018) and in recommender systems (Elahi et al., 2016). One of the major limitations is in the selection of the heuristic for data ranking and selection, whose performance can differ over different datasets. To address this shortcoming recent research has proposed methods for learning the selection heuristics themselves. One way of doing this is casting the problem as a reinforcement learning problem, where the learned policy takes the place of the predefined heuristics (Fang et al., 2017; Woodward & Finn, 2017; Pang et al., 2018). In the context of stream-based AL, the work of (Woodward & Finn, 2017) employs an LSTM to act as function approximator for a Q-network, and the output of the LSTM is connected to a fully connected linear layer producing the actual Q-values. The setup is very similar to the one in (Vinyals et al., 2016) where they address the problem in the few-shot learning setting. In (Pang et al., 2018; Bachman et al., 2017; Ravi & Larochelle, 2018), the process of learning the active learner is framed in a meta-learning setting. In (Pang et al., 2018) deep reinforcement learning is used to learn the active learning policy that generalize over different dataset, by using a generic embedding layers that maps dataset-dependent features to embeddings.

## 3 METHODOLOGY

In this section we describe the baseline model for active one-shot learning and the proposed extensions memory-augmented extensions. In particular we propose to augment the baseline with two different Memory Augmented Neural Networks (MANNs), the Neural Turing Machine (NTM) (Graves et al., 2014) and the Least Recently Used Access (LRUA) (Santoro et al., 2016a) memory.

### 3.1 PROPOSED MODELS

**LSTM Baseline Model**    The baseline, (Woodward & Finn, 2017), consists of a method for learning the active learner within a deep reinforcement learning framework. The model learn, with few examples per class, to make labelling decision online. The Q-function is approximated by an LSTM connected to a fully connected linear output layer (Figure 1a). The model trains on short episodes, in which it either predicts a class for an item received, or request the true label for it. Consequently, the output space of the model is $C+1$, where $C$ is the number of classes. The items for the given episode are randomly drawn from the training set, and are given a random slot in a one-hot vector indicating which class is associated with, *for the given episode*. In other words, the activation applied to the model will be episode specific, and should not force it to learn item-class binding dependencies. The number of items from every class in an episode vary, since the items are randomly drawn. Following the original work, we use Adam optimizer (Kingma & Ba, 2014) with default parameters, with the task of minimizing the Bellman error in the Q-network

**NTM-based Augmentation**    As the LSTM is relying solely on its internal state for representing the previous states, adding an external more explicit memory-structure could be helpful in increasing the accuracy of the system, similar to (Santoro et al., 2016b). We employed a Neural Turing Machine (NTM) as in (Graves et al., 2014) as the Q-network with an LSTM as memory controller (Figure 1b in the Appendix). As reported in (Santoro et al., 2016b), the NTM outperform the basic LSTM in a similar task setup, especially increasing accuracy on one-shot predictions Given that the NTM is a fully differentiable memory-structure, the model doesn't require a different task setup. For every episode, given the current state $h_t$, the LSTM controller produces an output which in turn is presented to all the read- and write-heads in the model. The read-heads returns a memory $r_t$ which together with the output from the controller, serves as input to the final fully connected layer producing the Q-values. It is important to note that the write-heads are *not* used when estimating the future discounted rewards - only the read-heads. This is because the model only *simulates* the next state and which Q-values it possibly would produce, and therefore shouldn't write anything to memory in this procedure.

**LRUA-based Augmentation**   The authors of (Santoro et al., 2016b) propose a different strategy for writing to memory using an NTM named LRUA (Least Recently Used Access). This strategy mainly differs in the writing-to-memory process. Instead of only using the read-weights to determine where to write to memory, several additional weight-vectors are introduced. The LRUA is a more specialized version of the NTM with a pure content-based memory writer, with two main choices when writing: 1) write to the *least* recently used memory location, 2) write to the *most* recently used memory location. The main difference between the two choices is that the former approach is resetting the memory location before writing, successfully replacing the memory, and the latter is *updating* the most recently used memory location with possibly more relevant information. In this way, important information is kept (i.e. information that has been used recently), as well as the memory is constantly updated with new information. Thus for our task setup, the inclusion of new classes will most likely be written to the currently least used slot, while samples of already existing slots will either update the most recently used slot (if the previous sample was of the same class), or be written to a least used slot.

## 3.2   EPISODE CONSTRUCTION BY CLASS MARGIN SAMPLING

To further improve sample efficiency and model performance, we introduce Class Margin Sampling (CMS). As opposed to standard margin sampling, CMS estimates the margin between $T > 1$ samples of the same class, for a specified number of classes (usually $C_{cms} = C \times 2$). In the context of a one-shot problem, with the added possibility to request a label instead of making a prediction, the standard margin sampling offers limited information. This is because the first sample in every episode shouldn't have considerable bias towards a specific class, which anyhow should be considered noise. By this particular design, all first-instance Q-values provide little but no information about the model, as we always want the model to execute a label request to maximize the expected reward. Thus, instead of calculating the smallest margin in a pool of samples, we change the method to better fit our task setup, using a pool of *classes*. The procedure starts by randomly drawing a specified number $C_{cms}$ of classes from the training set, which will act as the pool of classes. From each class, it draws $T$ samples which are processed, and fed to the model in sequence, meaning that all T samples from a class is used as an episode. This process is performed one class at a time, and then followed by a reset operation of both memory and hidden state. The margin for each drawn class is then calculated based on the sum of the *minimum* absolute[1] Q-values generated by the $T$ samples. Thus it's more likely that classes the model easily recognizes after the initial observation are *not* selected as a training sample. This procedure serves to reduce the likelihood of the following previously occurring problems during training:

1. If a sample's class is assigned the same random label multiple times, it starts creating inter-episode sample-class bindings, which can result in unfortunate class-biases.
2. The sample classes that are easily recognizable or distinguish themselves most from others, given the model's current parameters, don't provide optimal information gain during training, and thus can be rejected.

The first problem is addressed in (Santoro et al., 2016b). The authors argue that the NTM and LRUA overfits on the one-hot vector class-encodings, and propose a more robust encoding scheme for reducing this phenomenon. Our task structure is not compatible with a similar scheme, and we employ CMS to help reduce the likelihood of overfitting. The second problem is usually addressed by employing margin sampling, and is also the main reason for our use of CMS. By evaluating a pool of classes, CMS will select the sample classes that provide the most valuable information from the pool, given the models current parameters. CMS will thus in a certain sense select the most difficult samples to classify. Increasing the number of classes drawn $C_{cms}$ could potentially enhance performance further, but will also result in slower data collection, and thus finding an equilibrium will be beneficial. The sampling procedure will increase the training time, but at the same time enhance the generality of the models.

Since the models are trained by RL, any added bias in the task setup - e.g. conditioning the data sampling procedure - can be viewed as "unnatural" and potentially inhibit the exploration done by the model. We believe that since CMS consider the value of *all* output nodes, the inherent exploration in the model is still maintained.

---

[1]The absolute value of the Q-values are calculated *after* the maximum values are selected.

## 4 Results and Discussion

We first evaluate the average classification accuracy in the given task as well as the percentage of label requests for both, the baseline LSTM-based model (**LSTM**) and the proposed models with Reinforced-NTM (**NTM**) and Reinforced-LRUA (**LRUA**), with CMS and without. The models were trained on 100.000 episode batches from the training set, and then evaluated on 10.000 episode batches. A summary of such results for both training and test set is presented in Figure 2a and 2b in the Appendix. A detailed summary of the results is reported in Table 1.

| Model | Instance (% Correct) | | | | Instance (% Requested) | | | |
|---|---|---|---|---|---|---|---|---|
| | 1st | 2nd | 5th | 10th | 1st | 2nd | 5th | 10th |
| LSTM *(Baseline)* | *51.6* | *78.6* | *81.4* | *82.4* | *62.8* | *8.30* | *1.2* | *0.7* |
| NTM | 52.5 | 77.9 | *81.8* | 83.0 | 63.3 | 8.4 | 1.6 | 1.1 |
| LRUA | 58.0 | 79.2 | 81.8 | 83.2 | 62.3 | **6.9** | 0.7 | 0.4 |
| LSTM $C_{cms} = 2$ | 53.3 | 78.8 | 82.6 | 83.2 | 63.3 | 9.9 | 1.2 | 0.5 |
| NTM $C_{cms} = 2$ | 50.8 | 77.7 | 83.0 | 84.2 | 62.2 | 9.3 | 1.7 | 1.1 |
| LRUA $C_{cms} = 2$ | 63.7 | **79.4** | 82.6 | 83.7 | 63.9 | 8.9 | 0.8 | 0.5 |
| LSTM $C_{cms} = 3$ | 52.7 | 77.9 | **83.1** | 83.7 | **61.6** | 11.0 | 1.1 | 0.5 |
| NTM $C_{cms} = 3$ | 52.9 | 78.6 | 83.0 | **84.1** | 62.7 | 11.5 | 1.92 | 1.04 |
| LRUA $C_{cms} = 3$ | **69.1** | 78.4 | 82.2 | 83.1 | 64.9 | 11.7 | **0.6** | **0.3** |

Table 1: Class instance accuracies on test set. Accuracies are only calculated from predictions made.

For both baseline and proposed approaches, we observe a drop in the actual average accuracy values from the training set to the test set. In particular observe a drop of $7\%$ in LRUA-based model (Figure 2a, right side), due to an overfitting on the training set. This was an expected behavior, also reported in (Woodward & Finn, 2017), on the same task, by using LRUA as external memory for one-shot classification. For the LSTM and NTM, the drop in the average accuracy is instead between $2 - 3\%$ (Figure Figure 2a, left side and middle). In the comparison between the baseline model and the reinforced ones (NTM and LRUA), we observed that LRUA model has a tendency of requesting less labels than both the LSTM and NTM, outperforming both of them in both accuracy and percentage of requested labels. This can be explained by the capabilities of learning *more* meta-information about the episodic structure than the other models, that turns in a behavior characterized by: 1) requesting more first class-instances in episodes, 2) requesting less late class-instances in episodes. For example, we can notice that the reinforced-LRUA based models is requesting instance $62.3\%$ (similar to the baseline and the reinforced-NTM based) but with an accuracy of $58\%$. This trend is kept for late-class prediction ($2nd$, $5th$ and $10th$), but with less instance requested (i.e. $0.7\%$ at $5th$ instance against $1.2\%$ of the baseline) suggesting that the reinforced-LRUA based model is learning a better active learning strategy for zero-shot classification of images than the other models. In the same Table we reported also the results of further experiments with augmenting each model with CMS, either with $C_{cms} = C * 2$ or $C_{cms} = C * 3$, hereby written $C_{cms} = 2$ and $C_{cms} = 3$, and a margin time $T = 4$. We observe that augmenting the models with CMS increase the percentages of label requests done in general by the reinforced-NTM based model.

## 5 Conclusion

In this work we have proposed and test a memory-augmented model for deep one-shot active learning. This model intend to advance the capabilities of neural architectures for the data regime where data is scarce and the labeling has a non-negligible cost. The proposed model consists in a crafted combination of deep reinforcement learning for learning an active learning sampling heuristics with augmented memory networks to account for the necessary fast adaptability and information distilling necessary for one-shot learning. To improve the training process we introduce a modification of margin sampling, denominated Class Margin Sampling (CMS) in order to leverage to the known class information in the margin sampling process.

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

# A APPENDIX

## A.1 LSTM BASELINE AND NTM-AUGMENTED ARCHITECTURES

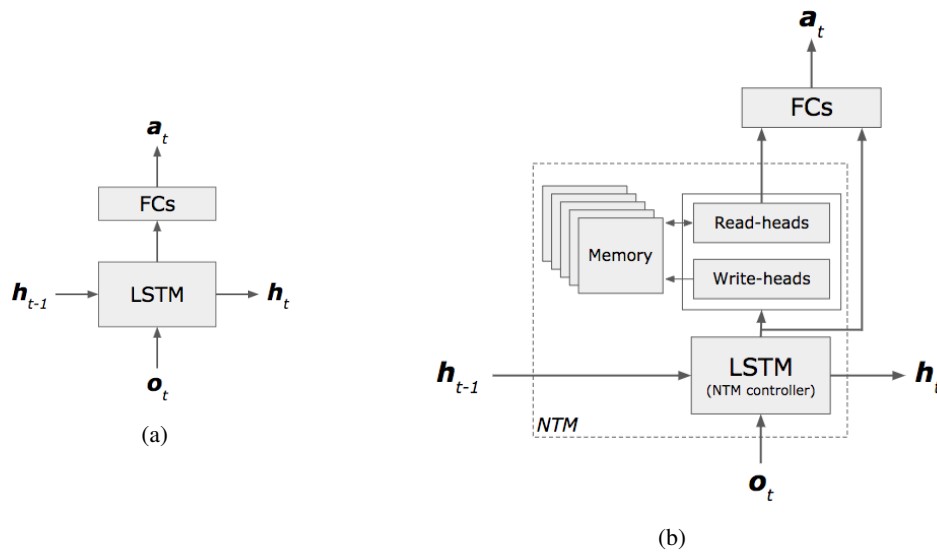

Figure 1: LSTM (Figure 1a) and NTM (figure 1b) architectures for active learning task

## A.2 EXPERIMENTAL SETTING

Our experiments have an episodic stream-based setup. Each episode is composed by 10 items for each class, with $C$ number of classes. The $C$ different classes are randomly sampled from the dataset before every episode and the 10 samples are randomly drawn for each of the $C$ classes. At the initial time-step in every episode, the model receives an example from the dataset, concatenated with a zero-vector of size equal to the number of classes $C$. The output space of the model can be divided into two choices: classify the example as one of $C$ possible classes or request the label of the example. As in (Woodward & Finn, 2017), we use a LSTM with 200 hidden units and a single hidden layer. The hidden layer is connected with a fully connected linear layer which outputs the Q-values. For training LSTM we use non-truncated BPTT (Back-Propagation Through Time). Both the input size and output size depend on the chosen number of classes per episode, with the notation $C$. The network has an input size of $20 \times 20 + C = 400 + C$, and the output size of the fully connected layer is $C + 1$, where the last node *always* represents the "request label" -action.

Additionally, the model employs an epsilon-greedy exploration strategy, with $\epsilon = 0.05$. If the model chooses to explore, there are three possible actions, each with $1/3$ probability. During the training process of the reinforcement learning agent, following (Woodward & Finn, 2017), we assign a reward of $r_t = -0.05$ for each label request, $r_t = 1.0$ for correct predicted label and $r_t = -1.0$ for wrong predicted label. We perform our test on a Image Classification task by using the Omniglot dataset (Lake et al., 2015), an image classification dataset consisting of 1623 classes of different characters from 50 different alphabets where each class consist of 20 hand-drawn characters. We preprocess the dataset by following the same procedure of (Vinyals et al., 2016). All the code of our experiments, for the sake of reproducibility is available on github [2].

## A.3 TRAINING AND TEST

---

[2]http:

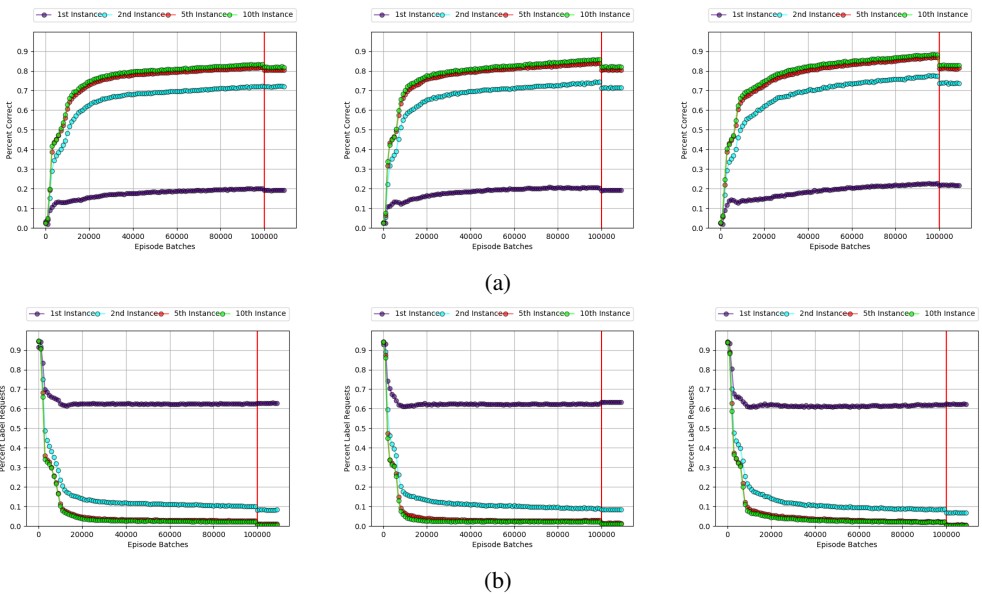

Figure 2: Average classification accuracy (Figure 2a) and request percentage (Figure 2b) for k-shot predictions in LSTM (left side), NTM (middle) and LRUA (right side). The red line indicates the switch between training and the test set for validation

