# OpenReview forum: "Augmented Memory Networks for Streaming-Based Active One-Shot Learning"
_ICLR.cc/2019/Workshop/LLD — Submitted to LLD 2019_

### Official Review · AnonReviewer2 · 2019-04-08
**Interesting idea, not reproducible experiments**

**Rating:** 1
**Confidence:** 2

**Review:**

This paper proposes bringing Augmented Memory Networks to streaming-based active learning, and experiments with 3 existing approaches of representing memory. Moreover, the authors propose an extension of margin based sampling, titled Class Margin Sampling (CMS).

While the idea of the paper sounds interesting and useful, this paper however is very difficult to read and is not reproducible. While I understand that the 4 page constraint limits how ample the explanations can be, there are some crucial details missing, that could have been added by simply citing other papers. I believe once this paper is rewritten it can make for an interesting contribution, but in its current form it requires a major revision.

Here are, in my opinion, the main weaknesses, in the order of importance:

1. Experiments: the experimental section reports accuracies on a dataset that is never mentioned. What is the task, how many classes are you classifying into? What is the input space? There is no description nor citation for this.

2. The paper is difficult to read, although I read it multiple times. Quite a few statements are not supported by explanations, which made it hard to follow the paper. For example:
- CMS is an extension to standard margin sampling, but standard margin sampling itself is never described or referenced. In order to understand why it needs improvement, we first have to understand how it works, even if it's a one sentence description.
- "adding an external more explicit memory-structure could be helpful in increasing the accuracy of the system, similar to (Santoro et al., 2016b)" -- so how is this work different than Santoro et al., 2016b?
- why is "usually C_cms = C × 2" ?
- "This is because the first sample in every episode shouldn’t have considerable bias towards a specific class, which anyhow should be considered noise."  -- why would it have such bias and why should it be considered noise anyhow?
- "followed by a reset operation of both memory and hidden state" -- no sharing of info between tasks/classes. Why?
- I don't understand problem 1 described on page 3.
- "Our task structure is not compatible with a similar scheme, "  -- why?

3. Editing and grammar issues:
- Forgetting fullstops at the end of sentences: e.g., "[...] increasing accuracy on one-shot predictions Given that the NTM [...]", also at the end of LSTM Baseline Model paragraph.
- Issues with subject and verb in a sentence: "all T samples from a class is used [...]", "Models that are capable of learning [...] is also of great interest".
- "Figure Figure 2a"

---

### Official Review · AnonReviewer3 · 2019-04-10
**Clear and well-executed workshop paper**

**Rating:** 3
**Confidence:** 2

**Review:**

This paper proposes a memory-augmented deep learning model for streaming-based one-shot active learning. It introduces Class Margin Sampling (CMS) which leverages known class information in the margin sampling process to improve sample efficiency in the context of active learning.

 CMS is well-motivated, explained, and is shown to improve results over baselines.  The experiments presented in Table 1 and their accompanying discussion provide a nice illustration of the work performed.

The paper demonstrates the use of active learning in three known architectures with and without CMS and provides a nice explanation of the models and why they were chosen.

Overall, the authors did a great job motivating and summarizing work that seems like a good fit for this workshop.  A longer version of the paper would benefit from additional discussion of the hyper parameters used in the RL agent and additional experiments.

Minor Suggestions:
- It might be helpful to mention the dataset used in Section 4 instead of just in the appendix.
- “learning policy that generalize over different dataset, by using a generic embedding layers that maps dataset-dependent features to embeddings.” -  plurality doesn’t seem consistent here.
- “By this particular design, all first-instance Q-values provide little but no information about the model, as we always want the model to execute a label request to maximize the expected reward” - wording
- “structure than the other models, that turns in a behavior characterized by:” - wording
- “In stream-based AL, observations are continuously made available to the learner that have to decide whether to request a label or to make a prediction.” -> “In stream-based AL, observations are continuously made available to the learner that has to decide whether to request a label or to make a prediction.”
- “The model learn, with few examples per class, to make labelling decision online” -> “The model learns, with few examples per class, to make labelling decision online”

---

### Official Review · AnonReviewer1 · 2019-04-10
**Better presentations and more experiments are needed**

**Rating:** 3
**Confidence:** 2

**Review:**

This paper presents a method for the stream-based active learning problem, which is a combination of active learning and one-shot learning.
The paper frame this problem as a reinforcement learning problem, and thus an AL-strategy (policy) is learned by an RL agent.
The main contribution of the proposed model is two-fold: equipping memory networks and a special sampling trick named Class Margin Sampling to deal with noisy initial samples.

The presentation of this paper is good. It would be very helpful to talk (with figures) about the overall workflow of the model at the beginning of Section 3 such that readers can have a big picture before they dive into the detailed solutions of each part.
A running example is also necessary to explain the whole stream-based AL problem and the solution proposed by the authors.

The two contributions are not particularly innovative, and the distinctions between the proposed work and (Pang et al. 2018) could be illustrated more.
The experiments on other datasets and comparisons with more baseline methods (even non-stram-based AL methods) are necessary to evaluate the proposed method.

I think this paper would become a strong paper after improving the presentations and experiments. But for now, I would say it is not that ready for publishing.

---

### Decision · Program_Chairs · 2019-04-16
**Acceptance Decision**

**Decision:**

Reject

**Comment:**

Reviewers appreciated the idea  but found it difficult to understand the details. Most notably major experimental details such as the datased are not described.